# Stimulus-Responsive Smart Nanoparticles-Based CRISPR-Cas Delivery for Therapeutic Genome Editing

**DOI:** 10.3390/ijms222011300

**Published:** 2021-10-19

**Authors:** Muhammad Naeem, Mubasher Zahir Hoque, Muhammad Ovais, Chanbasha Basheer, Irshad Ahmad

**Affiliations:** 1Department of Bioengineering, King Fahd University of Petroleum and Minerals (KFUPM), Dhahran 31261, Saudi Arabia; g201908410@kfupm.edu.sa (M.N.); g201805240@kfupm.edu.sa (M.Z.H.); 2National Center for Nanosciences and Nanotechnology (NCNST), Beijing 100190, China; movais@nanoctr.cn; 3Chemistry Department, King Fahd University of Petroleum and Minerals (KFUPM), Dhahran 31261, Saudi Arabia; cbasheer@kfupm.edu.sa; 4Interdisciplinary Research Center for Membranes and Water Security, King Fahd University of Petroleum and Minerals (KFUPM), Dhahran 31261, Saudi Arabia

**Keywords:** smart nanoparticles, CRISPR/Cas9 delivery, therapeutic genome editing, stimulus CRISPR delivery

## Abstract

The innovative research in genome editing domains such as CRISPR-Cas technology has enabled genetic engineers to manipulate the genomes of living organisms effectively in order to develop the next generation of therapeutic tools. This technique has started the new era of “genome surgery”. Despite these advances, the barriers of CRISPR-Cas9 techniques in clinical applications include efficient delivery of CRISPR/Cas9 and risk of off-target effects. Various types of viral and non-viral vectors are designed to deliver the CRISPR/Cas9 machinery into the desired cell. These methods still suffer difficulties such as immune response, lack of specificity, and efficiency. The extracellular and intracellular environments of cells and tissues differ in pH, redox species, enzyme activity, and light sensitivity. Recently, smart nanoparticles have been synthesized for CRISPR/Cas9 delivery to cells based on endogenous (pH, enzyme, redox specie, ATP) and exogenous (magnetic, ultrasound, temperature, light) stimulus signals. These methodologies can leverage genome editing through biological signals found within disease cells with less off-target effects. Here, we review the recent advances in stimulus-based smart nanoparticles to deliver the CRISPR/Cas9 machinery into the desired cell. This review article will provide extensive information to cautiously utilize smart nanoparticles for basic biomedical applications and therapeutic genome editing.

## 1. Introduction

Genome editing is an emerging field for precise manipulation of the genes in the genome of living organisms. In gene editing, programmable nucleases are engineered to target specific genes in the living organism at the genomic level. This is a prevailing domain to understand the nature of human genetics, genes and to develop therapeutics against genetic, cardiovascular, and cancer diseases (Table 1) [1]. Currently, researchers have developed the three types of gene editing tools including, Zinc Finger nucleases (ZFNs), Transcription Activator-like Nucleases (TALEN), and the latest developed Clustered Regularly Interspaced Short Palindromic Repeats (CRISPR). CRISPR is the latest emerging revolutionizing genome editing technology in the domain of biological sciences that is treating genetic diseases in health and improving crops against biotic and abiotic diseases in agriculture on a sustainable basis [2]. CRISPR has two components; Cas9, a nuclease that cuts the DNA at target loci and single guide RNA (sgRNA), which guides the Cas9 to create a cut at a particular or desired site of DNA. Programmable nucleases such as Cas9 cut specific DNA sequences followed by a trigger of DNA repair systems such as non-homologs end joining (NHEJ) and homologues direct repair (HDR). NHEJ leads to gene knockout (KO) and the HDR pathway leads to gene knock In (KI) with an additional component, 48 bp upstream and downstream homology arms [3]. Previous genome editing technologies such as ZFNs or TALEN rely on protein engineering of nucleases and causes off-target effects. In contrast, CRISPR is based on RNA-guided engineering technology, which directly targets DNA and shows relatively low off-target effects [4]. CRISPR/Cas9 is simple to use and has great potential to develop the next generation therapeutics. In the development of precise CRISPR technology, its delivery to target cells remains the main bottleneck. Different types of vectors, such as viral and non-viral, have been designed for the transfer of the CRISPR/Cas9 system to overcome the transfection issue [5]. 

The commonly used adeno-associated virus (AAVs) and lentivirus vectors are employed to transfer the gene-editing machinery into the cell; despite their many advantages, the immunogenic response and limited packaging facility impede their use in therapeutic gene editing. Moreover, the long duration of Cas9 expression inside the cells causes off-target effects that lead to unintended mutations at the genomic level. That is why a transient delivery-based non-viral vector has been developed that can circumvent the off-target effects [6]. 

In the past several years, synthetic non-viral nanotechnology-based vectors were designed to increase the efficacy of CRISPR/Cas9 delivery into specific tissues. The key delivery property of NPs is the endosome escape process [7,8]. In addition, regulated and stimulus-based CRISPR/Cas9 delivery to specific cells or tissues has great potential to reduce the off-target effects in future therapeutic gene manipulation applications. In this regard, researchers developed stimulus-based smart NPs to improve the delivery effectiveness of CRISPR/Cas9 genome editing system. Previous reviews have summarized the CRISPR/Cas9 delivery strategies [9,10,11,12,13]. However, in this review, we discuss the cutting-edge smart NPs developed to solve the delivery issues of CRISPR technology at the cellular level (Figure 1) [14]. The smart NPs can be designed based on endogenous signals including pH, redox, and ATP and exogenous signals including radiations, magnetic field, and ultrasound waves to control or regulate the CRISPR/Cas9 delivery to specific cells in vivo. The controlled or regulated CRISPR/Cas9 delivery at a particular location and time in the cell has decreased the off-target effects in therapeutic gene editing. This review aims to summarize the current advances in gene editing delivery based on smart NPs and their future potential in the development of next generation therapeutic tools.

## 2. Definition of Smart Nanoparticles for Gene Therapy

Nanotechnology is the latest interdisciplinary field in which materials are engineered at the nanoscale for various purposes. Nanodevices and nanomedicine are broad terms which have emerged since the birth of nanotechnology. Nanodevices are being used in myriad fields such as energy, electronics, chemistry, and molecular biology. The convergence of healthcare and nanotechnology led to the development of nanomedicine or the nanotherapeutic field. Nanotherapeutics have revolutionized certain areas of therapeutics such as imaging, faster diagnosis, drug/gene delivery, tissue regeneration, and precision medicines. Several nanoparticles (NPs) were used as nanomedicine to deliver the drugs/genes and proteins to target cells. Many NPs are already approved for clinical usage, and many are under trial. Smart NPs are an advanced form of stimulus-based NPs that can be employed to deliver the cargo in vivo specifically and efficiently [15]. 

Ideally, the vectors suitable for gene delivery should possess the following characteristics. First, the vectors are compact structures that can protect the molecules (gene/mRNA/protein) from degradation and undesired interactions with the biological environment. Second, the vectors are capable of overcoming the extracellular and intracellular barriers to transfer the molecules into the target cells, e.g., endosomal escape and localizing in the nucleus. Third, the vectors have little to no toxicity and avoid stimulating the immune system. Additionally, the vectors are biodegradable but are able to induce sustained expression with high transfection efficiency over a defined period. Most advanced designed smart NPs possess the above-mentioned properties of suitable vectors [16,17]. 

## 3. To Be or Not to Be, the Necessity of Using Smart NPs for the Delivery of CRISPR Components to Target Cells

There are two prominent methods (viral and non-viral) that can be used to deliver the CRISPR components to target cells. Viral delivery of CRISPR/Cas9 components was the most commonly used strategy in the last decade with high efficacy of gene editing results. However, there are limitations of viral delivery, such as random insertion in the host genome, which can cause oncogenesis, limited packaging capacity, inability of some viral vectors to transduce in non-dividing cells, and immune stimulation/rejection problems that can limit their in vivo therapeutic applications to treat human genetic diseases through CRISPR/Cas9. To overcome these limitations, the non-viral or physical delivery methods such as microinjection and electroporation can be employed, but the application of physical gene delivery methods are limited due to cell toxicity low cell specificity and viability effects [18]. 

Recently, researchers have focused on NPs, especially smart NPs to regulate stimulus-based CRISPR/Cas9 delivery to particular cells. Smart NPs are non-viral vectors having advantages over viral vectors such as minimal immune response, high packaging efficiency, controlled release via stimulus-response, designing flexibility, and in vivo target cell specificity with low toxicity [19]. The key limitation of smart NPs is their low delivery as compared to viral vector-based CRISPR/Cas9 delivery methods. However, the delivery efficiency of smart NPs can be increased by designing advance synthetic stimulus-based biomaterials/NPs [20]. 

**Table 1 ijms-22-11300-t001:** CRISPR-based current clinical trials.

Disease	Target Gene	Phase	Reference
Severe sickle cell disease	BCL11A gene	I/II	[21]
β-thalassemia	HBB	I	[22]
Advanced esophageal cancer	PD-1	II	[23]
Gastrointestinal epithelial cancer	CISH	II	[24]
Hematological malignancies	CCR5	Not available	[25]
Solid tumors	PD-1 and TCR Receptors	I	[26]
Leber congenital amaurosis (LCA)	Photoreceptors gene	I	[27]
Hereditary transthyretin amyloidosis (hATTR)	Transthyretin (TTR) gene	I	[28]

## 4. Endogenous Stimulus-Responsive NP for CRISPR/Cas9 Delivery 

Endogenous stimulus-responsive NPs exploit the endo-signals of a cell such as pH, redox, and ATP concentration to regulate the delivery of CRISPR/Cas9 to a particular cell in a determined manner. There are many types of endogenous signals that can be employed for the development of smart NPs for CRISPR/Cas9 delivery. The most recent smart NPs created to deliver the CRISPR components to particular cells are pH, redox, and ATP sensitive. 

### 4.1. PH-Responsive NPs 

The physiological environment of cells shows significant pH differences; for instance, it is seen that the extracellular environment of the tumor has low pH (acidic), but cytosol of cells have pH near to neutral [11,29]. The acidic or low pH environment of tumor cells occurs due to fast multiplication of cancerous cells which causes shortage of oxygen and consequently triggers lactic acid production due to anaerobic respiration rather than aerobic respiration with oxidative phosphorylation. This process is known as the Warburg effect [30]. The pH-responsive nanomaterials show many advantages; such as enhanced cellular uptake of CRISPR/Cas9 through proton sponge effects, which help in efficient gene delivery to the particular cells and less toxicity to targeted cells [31]. Polyethylene (PEI) is a cationic nanomaterial extensively used for controlled drug/protein delivery against the pH-responsive environment through the promotion of the proton sponge effect [32]. Any designed cancers therapeutic agent to target specific cancerous cells inside the body passes through three consecutive barriers, (1) blood circulation, (2) accumulation in tumor tissues, and (3) entry into the cancers cells. To overcome these barriers, the delivery systems should have different surface properties. In a recent study, the multistage delivery system (MDNP) with different surface properties was designed to deliver the CRISPR/Cas9 components specifically to cancer cells. The MDNP core structure contains CRISPR/dCas9 (pDNA) with phenylboronic acid (PBA)-modified low molecular weight polyethyleneimine (PEI–PBA). However, the shell is composed of 2,3-dimethylmaleic anhydride (DMMA)-modified poly(ethylene glycol)-b-polylysine (mPEG113-b-PLys100/DMMA) (Figure 2). During blood circulation, the MDNP maintains the core-shell structure. While entering the tumor cells, the acidic environment induces the decomposition of the DMMA group that triggers the detachment of polymeric shell from the core that leads to exposure of polyplex core to the cationic surface which increases the cancer cells accumulation. In addition, the cancer cells have a high level of surface sialylation, the PBA group of polyplex increases the internalization into cancer cells that trigger the detachment of polymeric shell from the core and causes endosomal disruption, which leads to the release of CRISPR/Cas9 (pDNA) inside the cytoplasm through proton sponge effect. With this MDNP strategy combined with epigenome editing, they have increased the expression of tumor suppressor gene miR534 in mice cells line to inhibit cancer cells proliferation as a cancer therapy [33]. 

Researchers designed the self-assembly of DNA into DNA nanoclews (NC-12) to investigate the genome editing specificity and efficiency by combining the gRNA with Cas9 to make gRNA/Cas9 complex. The gRNA/Cas9 complex can be self-assembled into DNA nanoclews. The gRNA/Cas9 complex loaded into the nanoclews was further coated with polyethylenimine (PEI),high density ionizable amine that makes the nanoparticle pH-responsive through promoting the proton sponge effect for endosome escape that releases the targeted machinery to the intracellular environment efficiently in pH-responsive areas. In this study, they targeted the EGFP that reduces the expression of proteins up to 36%. Moreover, these DNA nanoclews were introduced into cancers cells to knock out the EGFP gene expression up to 25% in mice cell lines with fewer off-target effects [11]. That strategy can be employed to target specific cancer cells through CRISPR/Cas9. The porous zeolite imidazole framework is a type of metal–organic framework (MOF) recently used to reduce the expression of EGFP gene in response to an acidic environment as a controlled gene delivery method. The authors encapsulated the Cas9 RNP into zeolitic imidazolate framework-8 (ZIF-8) in situ. They discovered that the ZIF-8/Cas9 complex NPs releases a low amount of CRISPR system (Cas9/gRNA) to the cell under a neutral environment but it releases up to 70% of Cas9/gRNA machinery to the desired cell in the acidic environment or low pH within 10 min (Figure 3) and reduces the expression of EGFP fluorescence protein expression level by 30% inside the cells [34]. 

In the past, lipids have been widely used to deliver the mRNA and proteins to target cells for multiple purposes [36]. In one study, ionizable lipid smart NPs were used to deliver the CRISPR/Cas9 components specifically and efficiently to target cells with high uptake efficiency; ionizable particles such as PEI were used to ionize the head of lipid NPs, which increases the pH-responsive endosome escape property. The combination of pH-responsive lipid NPs and viral gene delivery method, adeno-associated viral vector (AVV) employed to correct the genetic mutation of tyrosinemia genetic hereditary disease through gene CRISPR/Cas9 knock-in strategy. The lipid NPs/AVV/sgRNA/Cas9 complex with homology arms to trigger HDR pathway delivered efficiently to mouse models to treat tyrosinemia disease. The gene mutation associated with tyrosinemia genetic hereditary disease was corrected efficiently up to 6% with less off-target effects in the mouse cells lines [37]. It has been discovered that the enzyme serine protease proprotein convertase subtilisin/kexin type 9 (PCSK9) is a key regulator and modulator of lipoprotein cholesterol (LDL-C). The Pcsk9 and LDL-C interaction causes a decrease in the clearance of LDL-C from the blood that ultimately leads to accumulation of LDL-C that causes coronary heart diseases. A triple target strategy was developed through the combination of gold nanoparticles modified with HIV-transactivating transcriptor (TAT) peptides, nuclear localization signal (NLS) tagged Cas9 protein to target the nucleus and targeted sgRNA (Figure 4). The combined complex of gRNA/Cas9/GNPs was encapsulated into galactose cationic lipid to target the glycoprotein receptors. That complex targets in the three modes, galactose targets the receptor protein on hepatocytes, TAT peptides and NLS target the nucleus and sgRNA/Cas9 complex delivered to liver cells to knock out and reduce the expression of Pcsk9 gene that decreases the 30% LDL-C in the blood with no detectable off-target effects [38,39,40,41]. 

Currently, pH-responsive silica metal–organic framework (SMOF) hybrid NPs have been reported, which contains silica as well as ZIF. The SMOF-NPs showed excellent loading capacity, high stability, and efficient intracellular delivery of various payloads such as small hydrophilic drugs and proteins in addition to CRISPR/Cas9 machinery into the targeted cells. Particularly, the loading content and loading efficiency of RNP was 9.8 wt% and 97%, whereas for RNP + single-stranded oligonucleotide DNA (ssODN) was 9.5 wt% and 94%, respectively. These impressive qualities of the SMOF are due to the pH-controlled release and endosomal escape as a result of the proton sponge effect mediated by imidazole groups. In vivo genome editing with RNP-loaded SMOF was successfully induced in murine retinal pigment epithelium (RPE) via subretinal injection [42]. 

### 4.2. Redox-Responsive NPs 

Redox reactive cellular environment is an effective system for the stimulus-responsive cancer therapy [43,44]. The redox-responsive NPs have more benefits than pH due to intracellular Glutathione (GSH), which helps in gene or CRISPR/Cas9 complex direct delivery to the nucleus. The redox cellular environment is governed by NADP+/NADPH redox couple, due to which, the concentration of GSH inside the cell is higher than the outer environment [45,46]. Cancer cells have a higher concentration of reactive oxygen species than healthy cells or normal tissues [47]. Both ROS and GSH are employed to develop efficient delivery of CRISPR components into the cells through NPs such as biodegradable lipid NPs. In one study, a bio-reduceable lipid NP was created for the efficient transfection of CRISPR/Cas9 delivery to the cell nucleus [48]. The lipid NPs were developed through the combination of aliphatic amines and acrylamides. These NPs can self-assemble the Cas9 RNPs and gRNA cargo inside, and after entry into the cell, NPs can be decomposed by degradation through intracellular glutathione. In this study, lipid 8-014B was employed to deliver CRISPR machinery into the HEK291 cell in order to suppress or knock-out the EGFP gene expression that results in 70% gene editing efficiency in mammalian cell lines. (Figure 5). 

It has been discovered that bioreducible lipid NPs are the best vehicle to transfer the mRNA into the cell for genome editing [49]. There are many particles that accumulate at the disease site when transferred to cells and show less efficiency and effectiveness. A phenylboronic acid (PBA)-conjugated glutathione oxidation-responsive lipid NP was developed to improve redox-based efficiency. The NPs were named PBA-BADP, which can deliver the sgRNA efficiently to the target cells to edit at the genomic level. This PBA-BADP NPs can attach to salic acid (SA) receptors on cancer cells, which can efficiently transfer the mRNA to the tumoral cells to knock-out the gene expression of GFP. In this study, researchers discovered that there is a 2 to 3 percent higher efficient gene knock out of GFP protein phenotypes in the cancer cells through PBA-BADP lipid-based NPs delivery as compared to viral gene delivery of CRISPR machinery to the targeted cells [50]. 

The glutathione (GSH) redox-responsive nanocapsules are also developed to edit the genome in vivo and in vitro. These types of nanocapsules were formed through the polymerization of acrylate units, which assembled themselves around the Cas9 RNP. Moreover, imidazole compounds containing the disulfide are attached to the RNP complex through H-bond linkage. This imidazole helps in endosomal escape delivery of the redox-responsive NPs. These types of bioreducible nanocapsules employed to deliver the sgRNA/RNP complex to a targeted cell in vitro and in vivo to edit retinal pigment and skeletal muscles genes, respectively [51,52,53]. Recently, GSH-responsive silica NPs (SNP) reported with high loading content and loading efficiency. Here, the silica network was integrated with disulfide cross linkers and an imidazole-containing component was also included in the NP infrastructure. While the disulfide crosslinks bestow the NP with GSH-responsive payload release ability when taken up by a target cell, the imidazole group improves endosome escape capacity. The in vivo studies established the NP’s ability to effectively deliver sgRNA /RNP (intravenous injection) and mRNA (subretinal injection) to hepatocytes and murine RPE cells when the NPs are functionalized with GalNAc and all-trans-retinoic acid (ATRA), respectively leading to efficient genome editing. Figure 6 shows a schematic representation of synthesizing GSH-responsive SNPs conjugated with different targeting ligands (Figure 6) [54].

Initially, during the early days, it was difficult to transfect the Raw264.7 cancers cells and mesenchyme cells derived from bone marrow. Currently, black phosphorus nanosheets have been developed based on engineered positively charged nuclear signals such as Cas9N3 to enhance the electrostatic interaction. Black nano-phosphors are ROS responsive, which is automatically degraded in the ROS environment. This study suggests that CRISPR machinery can be delivered to mesenchyme and Raw264.7 cells efficiently with low off-target effects [55]. 

### 4.3. ATP-Responsive NPs 

Adenosine triphosphate (ATP) is the currency of the cell, which is the primary source of energy for living organisms. The amount of ATP in the intracellular environment is 1000 times higher than the outer cell environment [56,57]. That is why ATP concentration is a better choice to design the CRISPR/Cas9 machinery to target high and low ATP cells in living organisms. In one study, researchers designed the zinc-containing NPs, ZIF-90, to target the high ATP-containing cells inside the living organism. Due to competitive coordination of zinc ions and ATP that degraded the zinc atoms, the CRISPR/Cas9 system was released to the target cells. This whole mechanism has developed the ZIF-90 as an ATP-responsive NP [58]. The Cas9 can be encapsulated through self-assembly into the ZIF-90 metal–organic framework that can be easily transferred to targeted cells for specific and efficient therapeutic genome editing. In the presence of ATP, the NP releases the CRISPR/Cas9 cargo into the cell. The delivery of CRISPR/Cas9 machinery with gRNA through ZIF-90 decreased the genes expression of EGFP green fluorescence protein up to 40% in the HeLa-derived cells (Figure 7). It shows that the intracellular level of ATP can define the CRISPR/Cas9 delivery efficacy. This study opened the new domain of research to treat neurogenetic diseases, which are due to abnormal production of ATP, such as Alzheimer’s and schizophrenia. 

## 5. Exogenous Signals-Responsive Nanomaterial for CRISPR/Cas9 Delivery 

The usage of stimulus-responsive NPs to transfer CRISPR/Cas9 and their control through exogenous signals can improve the gene-editing efficiency in living organisms. In this section, we discuss different types of synthetic, organic, and inorganic NPs employed to transfer CRISPR/Cas9 machinery via external stimuli such as magnetic field, light, temperature, and ultrasounds. 

### 5.1. Photo-Responsive NPs 

Light-responsive CRISPR/Cas9 delivery into the targeted cells is an advanced remote-controlled CRISPR/Cas9 genome editing strategy in the tissues specific context of living organisms [59]. In previous studies, researchers developed gold NPs to deliver the CRISPR/Cas9 machinery into specific cells that were radiation- or light-responsive. The light-responsive gold NPs are due to surface plasma that creates the thermal effect that persuades the changes in the shape of NPs. The change in shape ultimately releases cargo machinery such as CRISPR components into the cell cytosol [60]. In another study, the gold NPs were coated with TAT peptides or amino acids that increased the encapsulation efficiency of CRISPR/Cas9 machinery into the gold NPs. With different analytical parameters, studies showed that 20 to 25 min of laser irradiation can increased the CRISPR/Cas9 complex release to target tissues and their editing efficacy was identified with a few off-target mutations. This gold NPs irradiation-based CRISPR delivery is termed, used to target the PLK1 gene, which is responsible of cancerous angiogenesis in the mouse, decreased the gene expression in tumor or cancerous cells both in vivo and in vitro [61,62]. 

The photo-liable semiconductor polymer NPs (pSPN) employed to transfer the CRISPR/Cas9 genome editing complexes to the targeted cells [63]. The NPs such as pSPN are light or irradiation responsive, which degrades itself and releases the payload (CRISPR components) inside [64]. The pSPN contains an oxygen-generated backbone and polyethyleneimine (PEI) brush. Light or laser light causes cleavage of the oxygen atom and PEI part from the pSPN that ultimately releases the CRISPR/Cas9 machinery into the desired target cells of the living organism. This irradiated or light-responsive gene-editing technique has increased the delivery of CRISPR/Cas9 machinery into mice cells by 15-fold compared to the non-irradiated CRISPR/Cas9 delivery [61]. Similarly, fabricated PEI functionalized carbon dots (CD-PEI) were designed for the intracellular delivery of CRISPR machinery. Carbon dots are photo-luminescent carbon-based NPs having high stability, biocompatibility, customizable optical properties and wide excitation spectra. The CD-PEI-fabricated NPs are suitable for delivery of plasmid CRISPR (pCRISPR) cargo owing to their size, shape, zeta potential, and one or more cellular uptake mechanisms. To study gene delivery capacity, HEK-293 cells were transfected with CD-PEI/pCRISPR nano-complexes encoding GFP as a reporter. The nano-complexes were reportedly internalized by more than 70% of the HEK-293 cells, depending on the weight ratios of CD-PEI to pCRISPR used. It was observed that a higher weight ratio (CD-PEI:pCRISPR of 100:1 and 200:1) gives the best results [65].

Recently, the researchers fused the heat-responsive promoter (HSP70) to regulate the genome-editing mechanism inside the cell [35]. This research showed that the radiation induces the photothermal effects in the nanocarrier of CRISPR/Cas9 machinery that regulate the expression level of Cas9 protein. This strategy can edit the cells from deep tissue of body cells with high resolution. The in vitro gene editing has remarkably created the INDELS or frameshift mutation to knock out the genes expression easily, effectively, and efficiently. This strategy can be used for target and treatment of deep tumors inside the body of mice and humans. 

The upconversion nano-carriers (UCNP) has been created, which can convert infrared radiation into ultraviolet light, which payloads the CRISPR/Cas9 and gRNA gene-editing machinery to target cells [66]. This study was performed to target the PLK1 gene responsible for cancer progression. The gene editing through UNCP NPs has reduced the gene expression level up to 45% that has reduced the growth of the cancerous cells up to 35%. Researchers discovered that this strategy can be useful to treat cancer cells through CRISPR/Cas9 gene-editing machinery [67]. Moreover, a charge-reversal nano-vector was designed based on an ultraviolet-sensitive conjugated polyelectrolyte coated on an upconversion nanomaterial (UCNP-UVP-P). The cationic side-chains (polyethylene glycol, UVP) of the nano-vector are converted to anion chains upon UV emission by the UCNPs following 980 nm light irradiation. Under near-infrared (NIR) photoirradiation, the charge conversion released plasmids from the nano-vector with 90% efficiently. Again, the target gene in the study was the PLK1 gene, whose up-regulation as a result of mutations is implicated in a variety of cancers. UCNP-UVP-Cas9/sgPLK-1 was used to treat proliferating HepG2 cells followed by 980 nm laser irradiation (Figure 8). The experimental results showed that the strategy was able to site specifically knock out the target PLK1 and consequently down-regulate PLK1 expression as well as induce apoptosis of the tumor cells. The transfection efficiency of tumor cells was reported to be ~63 ± 4% [68]. 

### 5.2. Magnetic Field-Responsive NPs 

With the advent of CRISPR/Cas9, the issue of off-target genetic modification arose, which still remains a serious unsolved problem of this technique [69]. It is also difficult to modulate the systematic delivery and replication of viral vectors that increases the risk of off-target effects and genotoxicity [70]. Recently, researchers developed a magnetic field-responsive CRISPR delivery technique by combining baculoviral vectors and magnetic-based NPs, which can modulate the gene expression through transient deliver/activation of CRISPR system in vivo by exposure of magnetic field (on or off) with less off-target effects and high specificity. In this case, the delivery of the CRISPR machinery can be regulated by applying a magnetic field to edit specific sites of the genome in various cells and tissues of living organisms [71,72,73]. The baculoviral vector expresses the gene without entering into the host cell (Figure 9). 

This CRISPR/Cas9 delivery technique has the potential for multiple genome editing in mammalian cells and tissues. Efficient delivery of non-viral CRISPR/Cas system using magnetic NPs (MNPs) reported in the literature. One such example is the fabrication of PEI-coated NPs having a magnetic iron oxide core. These NPs were utilized to make magnetic plasmid DNA lipoplexes. To test its efficiency, the CRISPR/Cas9 construct was prepared to target and knock-o t the porcine H11 locus in fibroblasts. The transfection method, in this case, was performed using magnetofection, which was reported to have 3.5 times more editing efficiency than the classical lipofection technique. The improved efficiency of magnetofection is credited to the magnetic properties of the NPs. When placed in a magnetic field, the NPs gain accelerated sedimentation on the cell surface, which also increased the concentration of the magnetic complexes on the surface of the cells. Additionally, non-specific interactions between the MNPs themselves and other serum components in the culture media were reduced, which resulted in higher cellular uptake. In this study, the researchers also reported the absence of any cytotoxicity associated with the MNPs [74]. Likewise, in a recent study, PEI-MNPs complexes with the CRISPR/Cas9 system were magnetofected into HEK-293 expressing traffic light reporter (TLR-3) system. The capacity of the magnetoplexes to deliver the payload and perform the HDR and NHEJ events were studied using a modified TLR-3 system containing an expression cassette of a non-functional GFP gene and a non-functional BFP gene. Depending on the efficiency of delivery, CRISPR/Cas9-mediated site-specific DSBs were corrected by either HDR (measured by GFP expression) or NHEJ (measured by BFP expression). In both studies, the researchers concluded that PEI-MNPs are promising as delivery vectors for plasmids encoding the CRISPR/Cas9 machinery.

### 5.3. Ultrasound-Responsive NPs 

Some nanomaterials can convert mechanical energy into motion that possess potential to be developed into controlled or regulated CRISPR/Cas9 machinery delivery methods to edit the gene in any specific cell. Thus, researchers have developed gold nanowires to transfer the CRISPR/Cas9 or Cas9 RNP complex to the targeted cells. The CRISPR cargo such as Cas9 RNP can self-assemble into the gold nanowire particle through a reversible disulfide bridge that can be released later into the cell cytosol through triggering the bridge degradation. It has been reported that, in 5 to 8 min of ultrasound treatment, the cells can enhance the delivery of CRISPR/Cas9 and gRNA complex to myeloma tissue of mouse model. It was shown that GFP gene expression has been reduced by 80% through knockout [75]. Recently, a microbubble–nanoliposomal particle has been developed as a carrier for Cas9/gRNA riboprotein complex; if stimulated by ultrasound waves, it can facilitate the effective delivery of the cargo at a specific site [76]. The researchers successfully transferred Cas9/sgRNA into the hair follicle dermal papilla cells of androgenic alopecia animals. Cas9/gRNA transfer followed by ultrasound activation causes recovery of hair growth by suppression of SRD5A2 protein production through the CRISPR gene editing system [76]. Overall, such ultrasound-responsive NPs shown great perspective in CRISPR/Cas9 delivery into different diseased cell types such as cancerous and syndrome cells. 

## 6. Dual-/Multistimuli-Responsive NPs

The notion of dual- or multistimuli-responsive NPs is not new. Varieties of NPs responding to more than one stimuli have been described previously for applications in anti-cancer drug delivery. Various combinations such as pH/magnetic field, pH/temperature, pH/redox, pH/magnetic/redox, and many other combinations have been reported for cancer therapy [77]. The responses of the NPs toward these stimuli are either simultaneous or they may be sequential in different compartments and/or environments. For instance, a mesoporous silica nanocarrier has been reported that was responsive to both pH and near-infrared radiation (NIR). The resulting nanocarriers loaded with cisplatin had selective uptake patterns that exhibited successful destruction of HeLa cells due to its chemo-photothermal therapeutic activity [78]. Similarly, a dual-responsive mesoporous silica-based polypeptide NP was designed for photo-thermal and photodynamic therapy. The thiol-modified polylysine NPs have been customized with disulfide bridges, PEG, and pH-sensitive dimethylmaleic anhydride, all of which renders the NP with pH- and reduction-responsive properties [79]. Numerous other dual-/multistimuli-responsive NPs have been reported for their application in cancer theranostics [80]. However, most of these earlier reported NPs were not particularly designed with CRISPR/Cas delivery in mind. Therefore, the obvious question arising is whether dual-/multistimuli-responsive NPs are applicable in CRISPR/Cas delivery and what advantages might they offer in contrast to NPs with single stimuli responsiveness discussed earlier. Multistimuli-responsive NPs offer greater spatiotemporal control of CRISPR/Cas genome editing machinery at target sites. There is limited literature published recently that provides evidence for the pros of designing multistimuli-responsive NPs for CRISPR/Cas delivery. Recently, the codelivery of CRISPR/Cas9 RNP along with anti-tumor photosensitizer chlorin e6 has been reported using a NIR- and reducing agent-responsive nitrilotriacetic acid-based NPs in a murine tumor model. In response to NIR irradiation, chlorin e6 generated ROS which mediated the lysosomal escape of the NPs. Cas9/sgRNA release in the cytoplasm was triggered by the reduction of the disulfide bond in the NP infrastructure. The gene-editing machinery was designed to target the antioxidant regulator gene *Nrf2*, thereby improving cancer cell sensitivity toward ROS. In contrast, gene editing failed within normal tissues without NIR irradiation as the CRISPR/Cas9 RNP underwent lysosomal degradation [81]. 

Currently, researchers have designed a lactose-derived CRISPR/Cas9 delivery vector capable of responding to a combination of two internal stimuli: asialoglycoprotein receptor (ASGPr) and reducing agent. The galactose residue of lactose is specifically recognized by ASGPr which is overexpressed on hepatocellular carcinoma cells. The delivery vector loaded with the *survivin* oncogene knockout machinery can locate the tumor via specific binding of galactose to the ASGPr followed by endocytosis of the delivery system. Once internalized, the disulfide bonds will be broken under a reducing environment, which will promote the release of the CRISPR/Cas9 [82]. Combining both internal and external stimulus and even multiple stimuli of one kind, i.e., either external or internal, can have advantages that depend on the intended application of the designed NP. Controlling external stimuli such as the discussed NIR can help dictate the spatial and temporal dimensions of when and where an NP will evade the lysosomes. Internal stimuli such as the ones described here facilitate efficient payload release. Overall, both studies mentioned here represent the superiority of multistimuli-responsive NPs for better spatiotemporal control of payload delivery at target locations and the potential for such NPs to be used for synergistic genome editing and drug delivery as well as decrease off-target effects. Therefore, we believe that in the near future, avenues of CRISPR/Cas delivery research must focus on designing NPs that will respond to various types of stimuli simultaneously, and at the same time, have high loading content and loading efficiency as well as precise and efficient delivery capability.

## 7. SORT Nanoparticles

Selective organ-targeting (SORT) NPs are a series of NPs designed recently for tissue-specific delivery of mRNA and CRISPR/Cas9 genome editing tool [19,83]. Basically, these NPs were fabricated by precisely optimizing the lipid content of traditional lipid NPs (LNP). The basic strategy in designing SORT NPs is to incorporate varying proportions of anionic (18BMP, 14PA, and 18PA), cationic (DDAB, EPC, and DOTAP) and ionizable cationic (5A2-SC8, DODAP, and C12-200) lipids into LNPs (C12-200 LNPs, mDLNP, and MC3 LNPs). Targeted delivery of mRNA and CRISPR/Cas9 are demonstrated in lung, spleen and hepatic tissues using SORT NPs [84]. The research group determined the tissue specificity of their designed SORT LNPs by intravenously injecting luciferase mRNA LNPs (18PA mDLNP and DOTAP mDLNP) and transfecting different organs in mice. These results showed that the organ-selective transfection by mRNA LNPs depended on the different proportions of DOTAP and 18PA used to fabricate the SORT NPs. Liver-specific transfection was achieved without any use of DOTAP, whereas 10–15% DOTAP and 50% DOTAP exhibited spleen- and lung-specific transfection. Conversely, 5–40% 18PA showed spleen-specific transfection, but no delivery of the mRNA was observed in lung and liver tissues [83]. Similarly, intravenous injection of SORT LNPs carrying Cas9 mRNA and sgRNA complexes specifically delivered the payload in the liver and lungs of tdTom reporter mice. The tdTom (Red) fluorescence was observed when the gene-editing machinery successfully knocked out the LoxP flanked stop cassette, which previously prevented the expression of tdTom protein. Red fluorescent gene-edited tissues were observed in the relevant organs treated with lung-, spleen- and liver-specific SORT LNPs [83]. It can be concluded that cationic SORT lipids can specifically target mRNA delivery to the spleen and lung, while anionic SORT lipids can control targeted mRNA delivery to the spleen. Ionizable SORT lipids aids to improve mRNA transfection efficiency in the liver [84]. Although the SORT technology is not particularly dependent on exogenous or endogenous stimulus, pre-adjusting the internal charge of the SORT LNPs can offer us a predictable and accurate way for CRISPR genome editing in specific organs. This is especially significant when, today, various systemic therapeutic targets remain inaccessible due to lack of RNP carriers and the unfeasibility of rationally designing a NP that can deliver the payload in that particular site. Perhaps in the near future, an outcome of further studies on SORT NPs will be the integration of stimulus response characteristics along with diverse organ specificity. What the scientific community will have then are NPs that can deliver CRISPR machinery to specific organs and release the cargo upon exposure to particular exogenous and/or endogenous stimuli.

## 8. Conclusions and Future Outlook

Despite the great potential of smart NPs to deliver CRISPR/Cas9 in the mammalian genome, its clinical applications are not fully realized due to off-target effects and imprecise delivery bottleneck. Viral vectors have high genome editing efficiency, but the packaging capacity of gRNA and Cas9 is low. Moreover, the long duration of the viral vector also causes off-target effects. Non-viral vectors have also been employed to transfer the gRNA and Cas9 as a plasmid or Cas9 RNP. The stimulus-based delivery of CRISPR/Cas9 machinery to specific cells can be a better alternative to viral-based CRISPR/Cas9 transfection methods to overcome targeted delivery issues. In the past several years, stimulus-responsive smart NPs proved as highly efficient genome editing delivery methods with fewer off-target effects. This technology gives us the advantage of regulating the CRISPR/Cas payload delivery among different types of cells. 

For forthcoming studies, novel CRISPR/Cas9 transfection methods that can distinguish between healthy and diseased cells are crucial for high throughput therapeutic genome editing. The smart NPs gene-editing machinery will remain inert until activation through endogenous or exogenous stimuli. Such stimulus-responsive delivery techniques are proven to be a new revolutionary gene-editing delivery strategy. The multifunctional-based smart NPs, which have both exogenous and endogenous signal-responsive properties, could have additional benefits over single stimulus-responsive delivery methods. Great effort is still needed in the development of a smart NPs-based CRISPR/Cas9 delivery approach that can efficiently edit the genome. Eventually, therapeutic genome editing using smart NPs can reach a new milestone by reaching clinical trials without undesirable effects. 

## Figures and Tables

**Figure 1 ijms-22-11300-f001:**
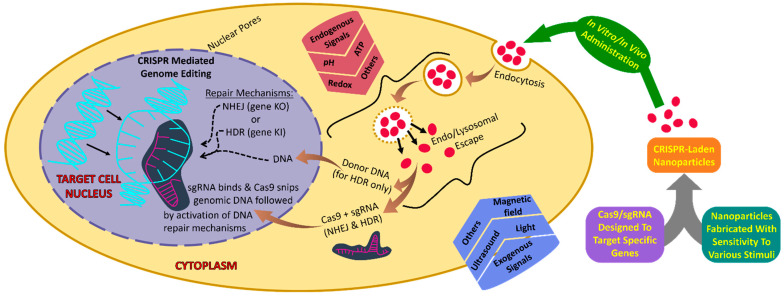
Schematic illustration of smart NPs stimulus-responsive-based CRISPR/Cas9 delivery.

**Figure 2 ijms-22-11300-f002:**
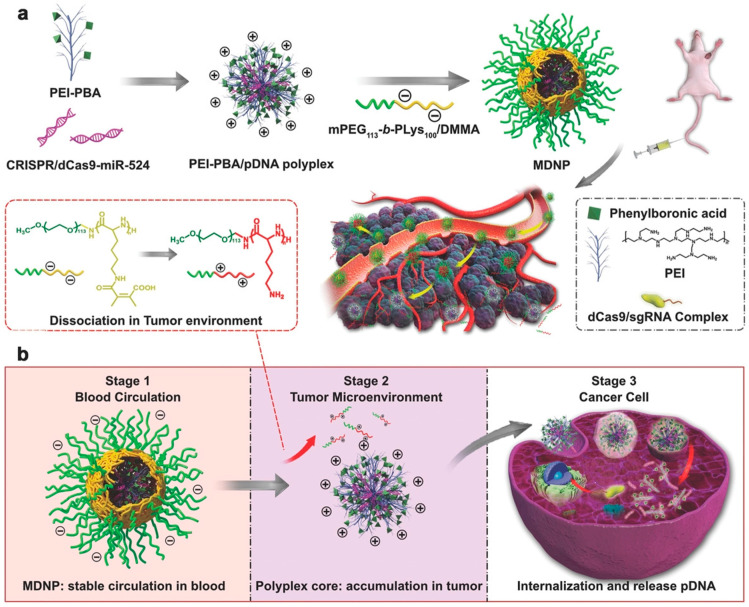
Schematic diagram of PEI/dCas9 to increase the gene expression of the miR524 gene as an anticancer therapy. (**a**) Preparation and delivery of MDNP. (**b**) MDNP delivery from blood circulation to the tumor cells. Reprinted with permission from ref [33]. Copyright 2018, Wiley-VCH.

**Figure 3 ijms-22-11300-f003:**
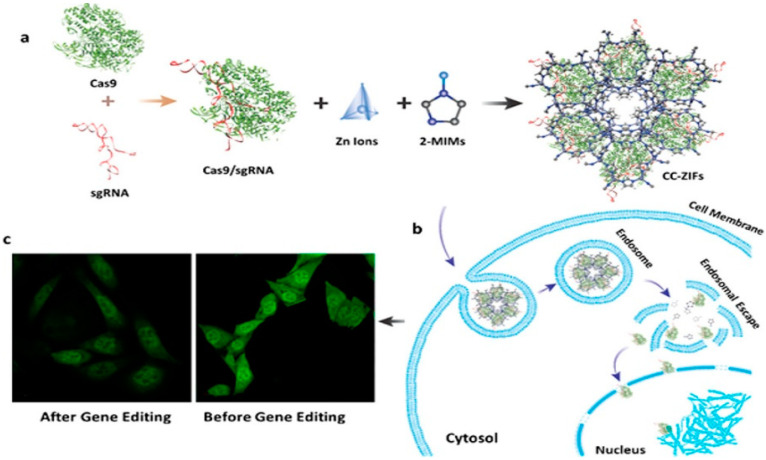
The gRNA/Cas9 complex assembled with the ZIF-8 metal–organic framework pores. The zif-8 was used to transfer CRISPR machinery to tumor cells efficiently. (**a**) Encapsulation of ZIF-8 nanoparticles. (**b**) Endosomal escape. (**c**) Image cell before and after treatment. Reprinted with permission from ref [35]. Copyright 2018, American Chemical Society.

**Figure 4 ijms-22-11300-f004:**
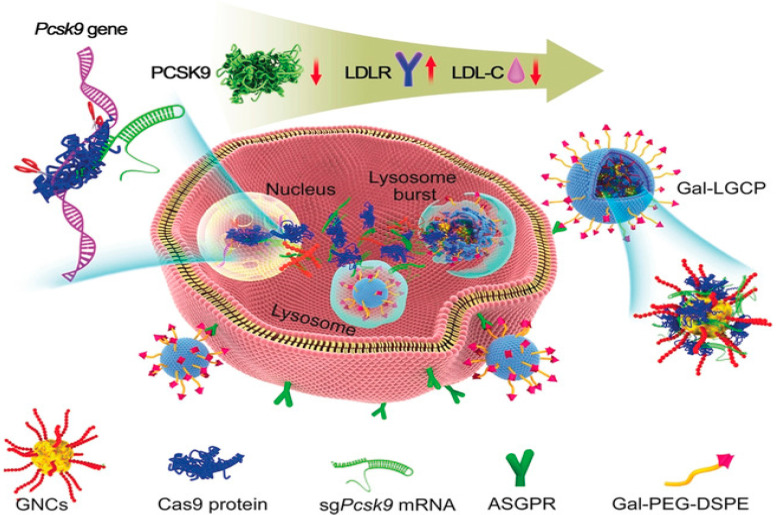
Gold NP coated with lipid for target delivery of CRISPR/Cas9. Reprinted with permission from ref [41]. Copyright 2019, John Wiley and Sons.

**Figure 5 ijms-22-11300-f005:**
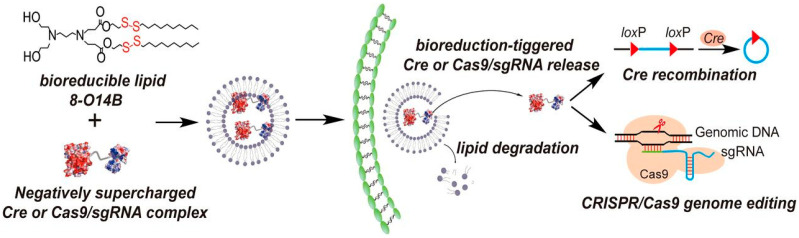
Glutathione-responsive bio-reduceable lipid NP for CRISPR/Cas9 delivery to induce the Cre recombination. Reprinted with permission from ref [48]. Copyright 2016, Proceedings of the National Academy of Sciences.

**Figure 6 ijms-22-11300-f006:**
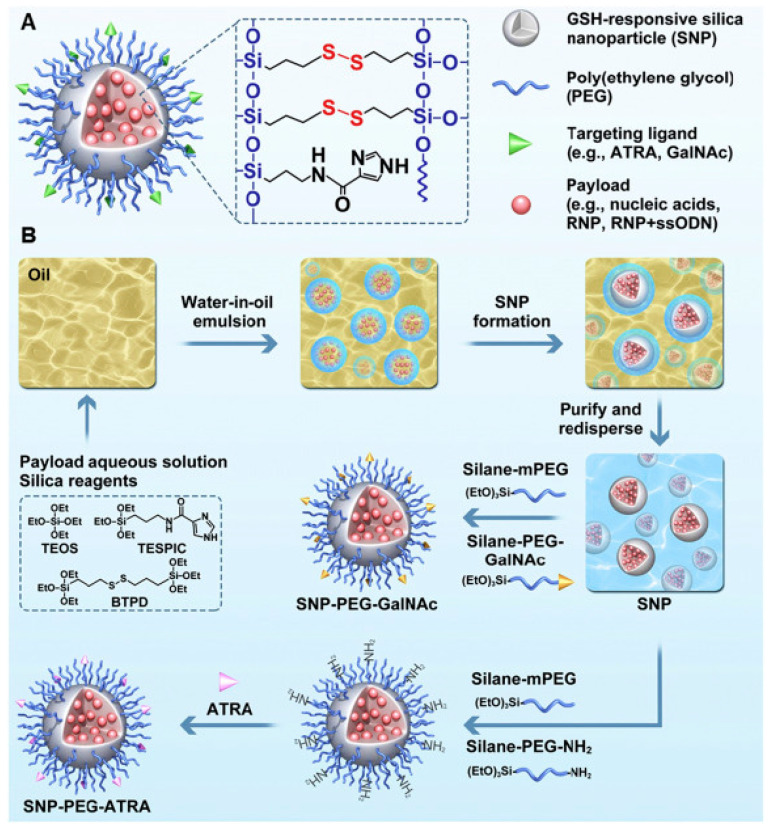
Design and synthesis strategy for multifunctional SNPs. (**A**) Representation of an SNP for nucleic acid and CRISPR components delivery. (**B**) Schematic for the synthesis of ATRA or GalNAc conjugated SNPs. Reprinted with permission from ref [54]. Copyright 2021, Elsevier.

**Figure 7 ijms-22-11300-f007:**
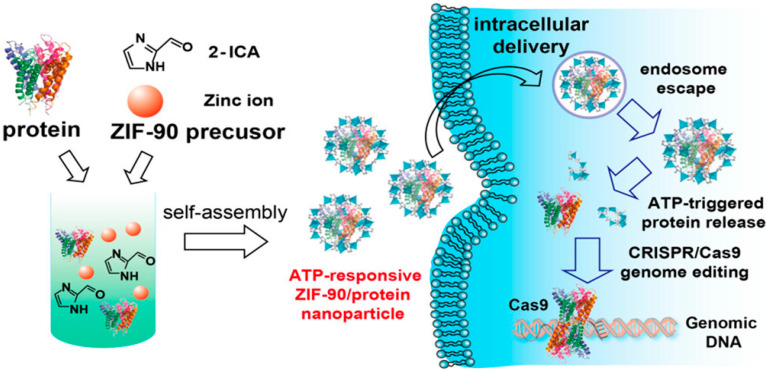
Metal–organic framework, ZIF-90 based ATP-responsive delivery of CRISPR/Cas9 machinery in the HeLa cells. Reprinted with permission from ref [58]. Copyright 2019, American Chemical Society.

**Figure 8 ijms-22-11300-f008:**
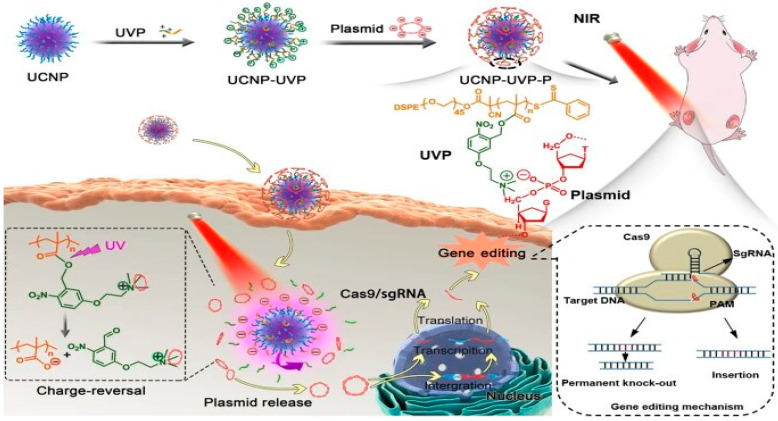
Schematic illustration of the synthesis of upconversion NPs with polyelectrolyte-coated (UNNP-UVP-P) and their usage in controlled delivery of CRISPR/Cas9 components. Reprinted with permission from ref [68]. Copyright 2020, Springer Nature.

**Figure 9 ijms-22-11300-f009:**
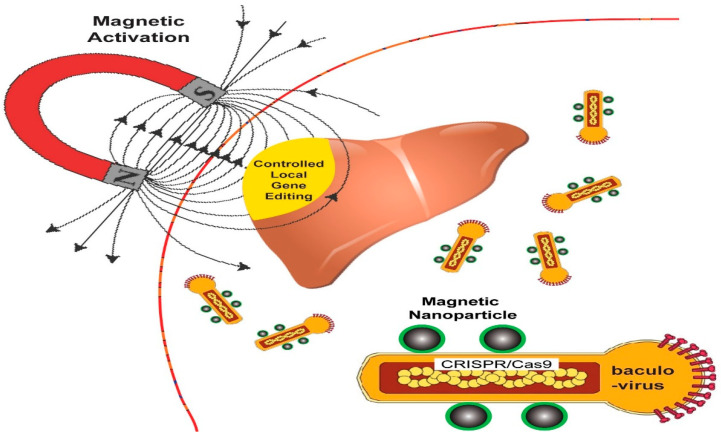
A schematic illustration of the MNP-BV system for in vivo delivery of CRISPR/Cas9 with magnetic. Reprinted with permission from ref [73]. Copyright 2018, Springer Nature.

## Data Availability

Not applicable.

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
