# Peer review of "Stimulus-Responsive Smart Nanoparticles-Based CRISPR-Cas Delivery for Therapeutic Genome Editing"

_ijms, 2021, doi:10.3390/ijms222011300_

Round 1
Reviewer 1 Report
Authors revise methods for delivering the CRISPR/Cas complexes to cells, in particular via inclusion in nanoparticles. The review comprehends most of the literature relevant for this topic and is well-organized. Critical comments are also present and discussed. This manuscript should be accepted with minor revision.
Minor English refinement is required.
Author Response
Response to Reviewer 1 Comments
Comment 1: Authors revise methods for delivering the CRISPR/Cas complexes to cells, in particular via inclusion in nanoparticles. The review comprehends most of the literature relevant for this topic and is well-organized. Critical comments are also present and discussed. This manuscript should be accepted with minor revision.
Response: Thank you very much for your comment and suggestion for the acceptance of our manuscript in the special issue of IJMS.
Comment 2: Minor English refinement is required.
Response: Thanks for your comment. We have carefully revised our manuscript and removed all minor grammatical mistakes.
Reviewer 2 Report
In this manuscript, the authors summarize different nanotechnologies available up to date to deliver CRISPR/CAS9 elements for genetic editing (gRNA, CAS9 RNP or mRNA) to target cells both in vivo and in vitro. In the first section, the authors introduce CRISPR/Cas9 technology and how it is applied on different gene editing strategies. Then they move on listing the currently used technologies (viral and non-viral-based strategies), their limitations, and the characteristics of an ideal smart nanoparticle (NP) for CRISPR/Cas9 delivery. Finally, they describe different types of smart nanoparticles for CRISPR/Cas9 delivery: Endogenous stimulus-responsive, Exogenous Signals-responsive, Dual-/Multi-Stimuli-Responsive, and SORT NPs.
For each category of delivery system, the authors provide multiple examples and a very complete outlook on the new technologies available. However, some sections are poorly written, sometimes it is even difficult understanding the meaning of the sentence. While the last chapters (6,7,and 8) are well written, chapter 4 and 5 should be heavily revised. In the first chapters (4 and 5) there are many typos and some sentences are impossible to follow. Moreover, the first chapters simply sum up the findings of different papers, sometimes even incorrectly (see ref 33). It would be beneficial to further summarize the findings of the different papers to focus more on the comparison and advantages/limitations of the different technologies.
Viral vectors are discussed as the alternative to smart NPs CRISPR/Cas9 delivery. More than once incorrect affirmations about such vectors are given, or comments about their specificity are made without a proper reference to back them up. This should be fixed.
Line by line corrections:
Line 59-60 “Morover, the long duration of viral vectors inside the cells causes off-target effect that leads to un-intended mutation at the genomic level [6]” I didn’t find any reference to this in the quoted review. Please clarify.
Line 111 “inability to transduce in non-dividing cells”. Lentiviral vectors can transduce non-dividing cells.
Line 147 substitute “to specific cancer cells” with “specifically to cancer cells”.
Line 151-4 Rephrase. In this way it is not clear the two-step dissociation described in the reference paper.
Line 153-4 “which leads to the release of CRISPR/Cas9 (pDNA) inside the cytoplasm to knockdown (KD) the miR534 expression.” Study in reference 33 do not perform such experiment, but the opposite. They use dCas9 to activate miR534 expression.
Line 157 Either say “tumor growth” or “the cancer cell proliferation”.
Line 171 Specify what ZIF-8 stands for.
Line 174 “that increases the pH-responsive endosome escape”. Rephrase. “to increase”, “in order to increase”, “which increases”
Line 187 substitute “heredity” with “hereditary”
Line 196-8 Rephrase. It is not immediately clear that the gold NPs were used to edit PCSK9 locus
Line 214. “provocation responsive” not clear
Line 233-4 “There are a lot of particles that accumu- 233 lated at the disease site when transferred to cells and show less efficiency and affectivity.”
Line 234. “affectivity” check the meaning of the word
Line 236. “effeminacy and affectivity” check the meaning of these words
Line 239 Substitute “tumorous” with “tumoral”
Line 241 Add references to the paragraph.
Line 243-260 References missing or not stated near relevant passage
Line 363-9 Not explained why the combination of baculoviral vectors and magnetic MPs should increase target specificity
Line 421 “at the same” is redundant
Line 504 “stimulus delivery bottleneck” Not clear what you mean
Line 508-10 “The stimulus-based delivery of CRISPR/Cas9 machinery 508 to specific cells can be proved as challenging for viral-based CRISPR/Cas9 transfection methods.” What? Please rephrase.
Author Response
Response to Reviewer 2 Comments
Major Points:
Comment: For each category of delivery system, the authors provide multiple examples and a very complete outlook on the new technologies available. However, some sections are poorly written, sometimes it is even difficult understanding the meaning of the sentence. While the last chapters (6,7,and 8) are well written, chapter 4 and 5 should be heavily revised. In the first chapters (4 and 5) there are many typos and some sentences are impossible to follow. Moreover, the first chapters simply sum up the findings of different papers, sometimes even incorrectly (see ref 33). It would be beneficial to further summarize the findings of the different papers to focus more on the comparison and advantages/limitations of the different technologies.
Response: Thank you very much for your comment. As following your guidance, we have extensively revised the chapter 4 and 5 of our manuscript. We have also removed the typos and grammatical errors.
Thanks for highlighting the mistake in reference 33; we have corrected it accordingly and checked all the references throughout manuscript.
Minor Points:
Comment 1: Morover, the long duration of viral vectors inside the cells causes off-target effect that leads to un-intended mutation at the genomic level [6]” I didn’t find any reference to this in the quoted review. Please clarify.
Response: Thanks for your comment. Sorry we misprinted it. Actually, it was long duration of Cas9 expression not viral vectors; we have changed it accordingly with reference at the line number 60, page number 2 of edited manuscript.
Comment 2: “inability to transduce in non-dividing cells”. Lentiviral vectors can transduce non-dividing cells.
Response: We have replaced the word “all” by “some” in lines number 112, 113 on page number 3 of the revised manuscript.
Comment 3: substitute “to specific cancer cells” with “specifically to cancer cells”.
Response: Thanks for your suggestion, we have changed the words “to specific cancer cells” by “specifically to cancer cells” in line number 153 on page number 4 of the revised manuscript.
Comment 4: Rephrase. In this way it is not clear the two-step dissociation described in the reference paper.
Response: Thanks for the nice suggestions; we have rephrased the mentioned paragraph from line number 148 to 167 on page number 4 of the revised manuscript.
Comment 5: which leads to the release of CRISPR/Cas9 (pDNA) inside the cytoplasm to knockdown (KD) the miR534 expression.” Study in reference 33 do not perform such experiment, but the opposite. They use dCas9 to activate miR534 expression.
Response: Thanks for your critical analysis and comment. We have changed it according to the reference study 33 from the line number 160 to 167 on page number 4 of the revised manuscript.
Comment 6: Either say “tumor growth” or “the cancer cell proliferation”.
Response: Thanks for your suggestion. We have changed it accordingly at the line number 150 on page number 4 of the revised manuscript.
Minor points:
Comment 7: Specify what ZIF-8 stands for.?
Response: Thanks for highlighting the mistake. We did the required correction in lines number 184, 185 on page number 5 of the revised manuscript.
Comment 8: “that increases the pH-responsive endosome escape”. Rephrase. “to increase”, “in order to increase”, “which increases” ?
Response: Thanks, we have edited it accordingly at the line number 197 on page number 6 of the revised manuscript.
Comment 9: substitute “heredity” with “hereditary”
Response: We have changed it accordingly at the line number 199 on page number 6 of the revised manuscript.
Comment 10: Rephrase. It is not immediately clear that the gold NPs were used to edit PCSK9 locus.
Response: Thanks for your comment. We have revised/edited the whole paragraph lines number 194 to 216 on page number 6 of the revised manuscript.
Comment 11: provocation responsive” not clear
Response: We have changed it from provocation to “stimulus” responsive in line number 230 on page number 7 of the revised manuscript.
Comment 12: There are a lot of particles that accumu- 233 lated at the disease site when transferred to cells and show less efficiency and affectivity.”
Line 234.“affectivity” check the meaning of the word
Response: Thank for highlighting the mistake. We have changed it accordingly in the line number 249 to 252 on page number 8 of the revised manuscript.
Comment 13: “effeminacy and affectivity” check the meaning of these words
Response: We have changed it accordingly in line number 252.
Comment 14: Substitute “tumorous” with “tumoral”
Response: We have changed the word “tumorous” by “tumoral” in the line number 255.
Comment 15: Add references to the paragraph.
Reference: Thanks, we have updated the reference at the line number 258 on page number 8 of the revised manuscript.
Comment 16: References missing or not stated near relevant passage
Reference: Thanks for highlighting the mistake; we have updated the reference at the line number 266 on page number 8 of the revised manuscript.
Comment 17: Not explained why the combination of baculoviral vectors and magnetic MPs should increase target specificity
Reference: Thanks for your comment. We have explained it the lines number 379 to 385 on page number 12 of the revised manuscript.
Comment 18: at the same” is redundant
Reference: Thanks, we have changed it accordingly.
Comment 19: stimulus delivery bottleneck” Not clear what you mean?
Response: Thanks for your comment; we have changed it accordingly from line number 523 to 524.
Comment 20: “The stimulus-based delivery of CRISPR/Cas9 machinery 508 to specific cells can be proved as challenging for viral-based CRISPR/Cas9 transfection methods.” What? Please rephrase.
Response: Thanks for your comment; we have rephrased it in lines number 527 to 529 on page number 16 of the revised manuscript.
Round 2
Reviewer 2 Report
No comment.